# Low-Energy Extracorporeal Shock Wave Ameliorates Streptozotocin Induced Diabetes and Promotes Pancreatic Beta Cells Regeneration in a Rat Model

**DOI:** 10.3390/ijms20194934

**Published:** 2019-10-05

**Authors:** Chang-Chun Hsiao, Cheng-Chan Lin, You-Syuan Hou, Jih-Yang Ko, Ching-Jen Wang

**Affiliations:** 1Graduate Institute of Clinical Medical Sciences, College of Medicine, Chang Gung University, Taoyuan 33302, Taiwan; poo779779@gmail.com; 2Center for Shockwave Medicine and Tissue Engineering, Kaohsiung Chang Gung Memorial Hospital and Chang Gung University College of Medicine, Kaohsiung 83301, Taiwan; s017066@gmail.com (C.-C.L.); kojy@cgmh.org.tw (J.-Y.K.); 3Department of Orthopedic Surgery, Kaohsiung Chang Gung Memorial Hospital and Chang Gung University College of Medicine, Kaohsiung 83301, Taiwan

**Keywords:** low-energy extracorporeal shock wave, streptozotocin induced diabetes, pancreatic beta cells regeneration, anti-inflammatory, anti-oxidative stress, tissue repair

## Abstract

Traditional therapy for diabetes mellitus has focused on supportive treatment, and is not significant in the promotion of pancreatic beta cells regeneration. We investigated the effect of low- energy extracorporeal shock wave (SW) on a streptozotocin induced diabetes (DM) rat model. Methods: The DM rats were treated with ten sessions of low-energy SW therapy (weekly for ten consecutive weeks) or left untreated. We assessed blood glucose, hemoglobin A1c (HbA1c), urine volume, pancreatic islets area, c-peptide, glucagon-like peptide 1 (GLP-1) and insulin production, beta cells number, pancreatic tissue inflammation, oxidative stress, apoptosis, angiogenesis, and stromal cell derived factor 1 (SDF-1) ten weeks after the completion of treatment. Results: The ten- week low-energy SW therapy regimen significantly reduced blood glucose, HbA1c, and urine volume as well as significantly enhancing pancreatic islets area, c-peptide, GLP-1, and insulin production in the rat model of DM. Moreover, low-energy SW therapy increased the beta cells number in DM rats. This was likely primarily attributed to the fact that low-energy SW therapy reduced pancreatic tissue inflammation, apoptosis, and oxidative stress as well as increasing angiogenesis, cell proliferation, and tissue repair potency. Conclusions: Low-energy SW therapy preserved pancreatic islets function in streptozotocin-induced DM. Low-energy SW therapy may serve as a novel noninvasive and effective treatment of DM.

## 1. Introduction

Diabetes mellitus is a worldwide major health problem. Diabetes mellitus is characterized by high blood sugar levels over a prolonged period [1]. The classic symptoms of untreated diabetes mellitus are unintended weight loss, increased urination, increased thirst, and increased hunger [2]. The key pathogenesis of diabetes mellitus is inflammation [1]. The pro-inflammatory cytokines interleukin-1β (IΛ-1β) and tumor necrosis factor-α (TNFα) contribute to beta cells apoptosis [3]. Oxidative stress also induces insulin-producing beta cells apoptosis [4,5]. Beta cells dysfunction or apoptosis plays an initial role in the development and progression of hyperglycemia [6]. Oxidative stress plays an important role in the development of microvascular complications in diabetes mellitus [7]. Uncontrolled hyperglycemia in patients with diabetes mellitus can lead to a host of microvascular complications including retinopathy, diabetic nephropathy, and neuropathy [8]. Therefore, restoring beta cell function is the overarching goal for improved therapy of diabetes mellitus. Adult pancreatic islets are constituted of 70% insulin-producing beta cells [9]. Glucagon-like peptide 1 (GLP-1) stimulates the differentiation of beta cells from ductal progenitor cells [10]. The GLP-1 can increase beta cells regeneration by promoting alpha cells to beta cells trans differentiation [11]. The Nkx6.1 can maintain the functional state of beta cells and act as the marker of pancreatic beta cell identity [12,13].

An extracorporeal shock wave is a longitudinal acoustic wave, traveling at the speed of ultrasound waves in water through the body tissues [14]. It is a single pressure pulse with a short needle-like positive spike and followed by a tensile part of several microseconds at a lower amplitude [14,15]. A low-energy extracorporeal shock wave (SW) exerts its effects through other mechanisms, such as increased angiogenesis [16], increased cell proliferation [17], anti-inflammation [17,18,19,20], inhibition of oxidative stress [17,20], anti-apoptotic [17], enhanced podocyte regeneration [17], activating axonal regeneration [21], improved nerve regeneration [22,23], and enhanced tissue repair [17]. Low-energy SW might inhibit inflammation by polarizing M1 macrophages to M2 macrophages [17,24]. Given that low-energy SW exerts anti-inflammatory, increased cell proliferation, anti- apoptotic, enhanced tissue repair, increased angiogenesis, and inhibited oxidative stress, low- energy SW may benefit streptozotocin induced diabetes (DM) pancreatic islets. However, whether low-energy SW can alleviate DM and promote beta cells regeneration is unknown. Therefore, we hypothesized that low-energy SW would enhance anti-inflammation, cell proliferation, tissue repair, anti-apoptotic and inhibit oxidative stress effects to restore the insulin-producing beta cells function to the normal level in the DM rat model by promoting beta cells regeneration.

## 2. Results

### 2.1. Low-Energy SW Significantly Reduced Blood Glucose, Hemoglobin A1c and Urine Volume in DM

Diabetes mellitus is characterized by high blood glucose levels and hemoglobin A1c (HbA1c) over a prolonged period [1,6]. HbA1c is the average level of blood glucose over the previous 2–3 months [25]. The low-energy SW had significantly reduced blood glucose level compared with the DM group (Figure 1a). The DM group had significantly increased blood HbA1c level compared with the normal control rats (Figure 1b). The low-energy SW had significantly reduced HbA1c level compared with the DM group (Figure 1b). The classic symptom of untreated diabetes mellitus is unintended increased urination [2]. The low-energy SW had significantly decreased urine volume level compared with the DM group (Figure 1c). Low-energy SW ameliorated classic symptoms in DM.

### 2.2. Low-Energy SW Significantly Enhanced Pancreatic Islets Area, c-Peptide, GLP-1 and Insulin Production

Pancreatic islets are constituted of 70% insulin-producing beta cells [9]. Hematoxylin and eosin (HE) staining demonstrated that the DM group had decreased pancreatic islets area compared with the normal control group (Figure 2a), whereas low-energy SW enlarged pancreatic islets area compared with the DM group (Figure 2a). C-peptide is used as a marker of insulin release and a measure of beta cells activity [26]. Immunohistochemistry (IHC) C-peptide staining revealed that the DM group had significantly reduced insulin release compared with the normal control group (Figure 2a,b), however, low-energy SW had significantly increased insulin release in the pancreatic islets compared with the DM group (Figure 2a,b). GLP-1 can increase beta cells regeneration by promoting alpha cells to beta cells trans differentiation [11]. GLP-1 staining revealed significantly diminished beta cells regeneration in the DM pancreatic islets which was ameliorated by low-energy SW (Figure 2c,d). The DM group had significantly reduced insulin level in the pancreatic tissue compared with normal control group; however, low-energy SW had significantly increased insulin level in the pancreatic tissue compared with the DM group (Figure 2e–g). Low-energy SW improved beta cells function in DM.

### 2.3. Low-Energy SW Enhanced Beta Cells Regeneration

DM is associated with pancreatic beta cells apoptosis and loss [27]. Immunofluorescence (IF) proliferating cell nuclear antigen (PCNA) staining and Western blotting analyses revealed that the low-energy SW had significantly increased cell proliferation in the pancreatic tissue compared with the DM group (Figure 3a–d). The marker of pancreatic beta cells is Nkx6.1 [12,13]. The DM group had a significantly reduced number of beta cells in the pancreatic tissue compared with the normal control group; however, low-energy SW significantly increased the number of beta cells in the pancreatic tissue compared with the DM group (Figure 3e,f), suggesting that low-energy SW enhanced beta cells regeneration potency.

### 2.4. Low-Energy SW Enhanced Tissue Repair and Angiogenesis

The stromal cell derived factor 1 (SDF-1) is a tissue repair marker [28,29]. The low-energy SW had a significantly increased level of tissue repair marker SDF-1 in the pancreatic tissue compared with the DM group (Figure 4a), suggesting that low-energy SW enhanced tissue repair potency. The pancreatic islets are richly vascularized. The vascular endothelial growth factor (VEGF) is an angiogenesis marker [30]. The DM group had significantly reduced angiogenesis in the pancreatic tissue compared with the normal control group, whereas low-energy SW had significantly increased angiogenesis in the pancreatic tissue compared with the DM group (Figure 4b). The low-energy SW significantly enhanced angiogenesis.

### 2.5. Low-Energy SW Alleviated Oxidative Stress and Cell Apoptosis

Oxidative stress can induce pancreatic beta cells apoptosis [4]. The compound 8- hydroxy- 2’- deoxyguanosine (8-OHdG) is a critical biomarker of oxidative stress [31]. Pancreatic tissue IHC 8-OHdG staining and urine 8-OHdG analysis revealed significantly increased oxidative stress in the DM group which was ameliorated by low-energy SW (Figure 5a–c). Apoptotic cell death was determined using terminal deoxynucleotidyl transferase dUTP nick end labeling (TUNEL) staining and cleaved Caspase-3 analysis. The DM group had significantly increased cell apoptosis in the pancreatic tissue compared with the normal control group; however, low-energy SW ameliorated cell apoptosis in the DM pancreatic tissue (Figure 5d–g).

### 2.6. Low-Energy SW Prevented Diabetes-Induced Pancreatic Tissue Inflammation

Inflammation is a crucial factor in the pathogenesis of diabetes mellitus [1]. The inflammatory mediators IL-1β and TNFα contribute to beta cells dysfunction and apoptosis [3]. The F4/80 is a major macrophage marker [32]. The marker of anti-inflammatory M2 macrophages is CD206 [32]. The DM group exhibited significantly high levels of IL-6, TNFα, and IL-1β in pancreatic tissue compared with the normal control group, whereas low-energy SW significantly reduced the levels of these inflammatory mediators (Figure 6a–c). Moreover, low-energy SW had significantly increased anti- inflammatory M2 macrophages and anti-inflammatory mediator IL-10 in the pancreatic tissue compared with the DM group (Figure 6d,e). Low-energy SW prevented diabetes-induced pancreatic tissue inflammation.

## 3. Discussion

Low-energy SW reduced blood glucose as well as decreased HbA1c and urine excretion in the rat model of DM. Moreover, low-energy SW increased beta cells number and insulin production in DM rats. This was likely attributed to the fact that low-energy SW decreased pancreatic tissue oxidative stress, inflammation, and apoptosis as well as increasing anti-inflammatory, tissue repair potency, angiogenesis, and cell proliferation. We also found that low-energy SW can alleviate beta cells loss which may contribute to the preservation of pancreatic islets function. Low-energy SW is a novel noninvasive and effective treatment of DM.

IL-1β and TNFα contribute to beta cells apoptosis [3,4]. Low-energy SW significantly decreased the proinflammatory molecules IL-6, TNFα, and IL-1β but increased the anti-inflammatory mediators IL-10 which may contribute to the preservation of pancreatic beta cells function. This study showed that low-energy SW increased anti-inflammatory M2 macrophages. Low-energy SW is also reported to enhance anti-inflammatory M2 macrophage infiltration in a rat model of diabetic nephropathy [17] as well as acute myocardial infarction [24]. Low-energy SW might inhibit inflammation by polarizing pro-inflammatory M1 macrophages to anti-inflammatory M2 macrophages. Low-energy SW has an anti-inflammatory effect against various diseases [17,29,33].

Beta cells apoptosis or dysfunction is a key factor in the pathogenesis of DM [27]. This study demonstrated that low-energy SW significantly reduced apoptosis markers cleaved-caspase 3, and TUNEL but enhanced beta cells marker Nkx6.1, cell proliferation marker PCNA, tissue repair marker SDF-1 and GLP-1, which may contribute to increase beta cells regeneration potency. Interestingly, our previous study also demonstrated that low-energy SW can enhance podocyte regeneration [17] and activate axonal regeneration [21].

Low-energy SW produces mechanical forces and leads to biological effects [17,28,34]; after that, upregulation of angiogenesis markers vWF and VEGF [17,28,29,35,36], the tissue repair marker SDF-1 [17,29,35] and decrease of oxidative stress [17,20,28] may thereby promote pancreatic beta cells regeneration and tissue repair. Low-energy SW is also effective in ameliorating renal dysfunction in a rat model of acute kidney injury, through stimulation of VEGF expression [37]. Low-energy SW can improve peripheral nerve regeneration [23].

The underlying mechanisms in the therapeutic effect of low-energy SW treating of various diseases have been studied, including enhancement of tissue repair [17,28] and angiogenesis [16,28,38,39] as well as suppression of inflammation [17,18,19,20] and oxidative stress [17,20,28]. In this study, even though extensive work was done to elucidate the therapeutic effect of low-energy SW treating of DM, the exact underlying mechanisms remain unclear. The proposed mechanisms underlying the therapeutic effect of low-energy SW treating of DM based on our findings are summarized (Figure 7).

## 4. Materials and Methods

### 4.1. Animals

The experimental animal studies were approved by the Institutional Animal Care and Use Committee of Chang Gung Memorial Hospital (Approval No. 2012050904), the project identification date of approval (16 January 2013). Male Wistar rats weighing 250–300 g were used for the experiment. Male Wistar rats were purchased from BioLASCO (Taipei, Taiwan) and were maintained in the animal center at the Kaohsiung Chang Gung Memorial Hospital under strict accordance with the principles and guidelines of the Association for the Assessment and Accreditation of Laboratory Animal Care (AAALC) for the care and use of experimental animals. Thirty-six rats were randomized to the normal control (Normal) group (*n* = 12), the streptozotocin induced diabetes (DM) group (*n* = 12) and the DM with low-energy extracorporeal shock wave (SW) (DM + Shockwave) group (*n* = 12).

### 4.2. Establishment of the DM Rat Model

Male Wistar rats were given a single intraperitoneal injection of 50 mg/kg streptozotocin (STZ) (Sigma-Aldrich, St. Louis, MO, USA) to induce diabetes as described previously [40,41]. The Normal group rats were injected only with citric acid buffer. Blood glucose level was examined one day after the STZ injection under fasting for 8 hours using a contour glucose meter (Freestyle, Abbott Laboratories, Alameda, CA, USA) via tail puncture. All STZ treated rats exhibited blood glucose levels above 250 mg/dL and were included in the study.

### 4.3. Low-Energy SW Treatment.

The DM rats were anesthetized by inhalation of 2.0% isoflurane, then placed on a warming pad for low-energy SW treatment. Ultrasonic gel was applied to the abdomen after shaving the rat’s hair. Ultrasound (Toshiba, Tokyo, Japan) was used, scanned to locate the location of the rat pancreas, and then a mark was made. Based on our previous study [17,21], the shock wave applicator (EvoTron R05, High Medical Technologies, Switzerland) with probe was placed on the mark (pancreas) and a total of 200 shocks was delivered at an energy density of 0.13 mJ/mm^2^ at a frequency of 200 pulses per min [17,21]. The DM rats received low-energy SW treatment once a week for ten weeks [17] (Appendix A).

### 4.4. Measurement of Urine Volume and 8-Hydroxy-2’-Deoxyguanosine (8-OHdG)

Urine excretion of each rat was collected using a metabolic cage. We collected 16 hour urine samples from the Normal, DM, and DM with Shockwave groups. The 8-OHdG level were determined by 8-OHdG ELISA kit (Abcam, Cambridge, UK) according to the manufacturer’s protocol.

### 4.5. Hematoxylin and Eosin (HE) Stain, Immunohistochemistry, and Immunofluorescence

Fresh pancreatic tissues were fixed in 4% PBS-buffered formaldehyde and embedded in paraffin under RNase-free conditions. Specimens were sliced longitudinally into 4 mm thick sections. Sections were dewaxed for staining with HE according to the manufacturer’s instructions (Abcam, Cambridge, UK).

For immunohistochemistry, unstained paraffin slides were baked at 56 °C overnight and deparaffinized in xylene solution twice, then rehydrated sequentially in 95%, 75%, and 40% ethanol and washed with 1 × PBS. Slides were cooked for 20 minutes in 1× antigen retrieval buffer (H3300, Vector Laboratories, Inc., Burlingame, CA, USA), followed by 3 rinses with 1× PBS. Slides were quenched with 1% hydrogen peroxide for 10 minutes before incubating with blocking solution (4% normal horse serum or goat serum in PBS with 0.1% Triton X). After that, sections were incubated with primary antibody diluted in blocking solution overnight at 4 °C. Primary antibody anti-C-peptide (Abcam, Cambridge, UK), anti-Glucagon-like peptide 1 (GLP-1) (Abcam, Cambridge, UK), or anti- 8- OHdG (Abcam, Cambridge, UK). Slides were then washed with 1 × PBS for 5 times and incubated with biotinylated secondary antibody (Vector Laboratories; dilution 1:150) in blocking solution for 1 h at room temperature. After washing 5 times with 1 × PBS, the slides were incubated in a Vectastain Elite ABC kit (Vector Laboratories ABC kit; PK-6100) for 30 min at room temperature. After washing 5 times with 1 × PBS, slides were processed for color reaction with peroxidase treatment with the 3,3’-diaminobenzidine (DAB) substrate kit (Vector Laboratories; SK-4100), washed with tap water, and counterstained with hematoxylin. Ten pancreatic islets from three slides that were obtained from each rat were randomly selected for Carl Zeiss Axioskop 2 plus microscope (Carl Zeiss, Germany). Three random images from each selected region then were taken, captured, and analyzed using image analysis software (Image-Pro Plus software, Media Cybernetics, Silver Spring, MD). The positive immunolabeled cells and total cells in each area were counted, and the percentage of positive labeled cells was calculated.

For immunofluorescence, sections were rinsed in 1 × PBS after deparaffinization, and incubated for 1 h at room temperature in blocking solution containing 4% normal goat or horse serum in 1 × PBS with 0.1% Triton X. Primary antibodies were diluted in blocking solution and incubated overnight at 4 °C. Primary antibodies Insulin (Abcam, Cambridge, UK), anti-PCNA (Abcam, Cambridge, UK), Nkx6.1 (Invitrogen, Carlsbad, CA), anti-SDF-1 (Santa Cruz, Dallas, Texas), vWF (Abcam, Cambridge, UK), CD206 (Abcam, Cambridge, UK), or F4/80 (Santa Cruz, Dallas, Texas). Subsequently, sections were rinsed in 1 × PBS three times and incubated with suitable fluorescent secondary antibodies (Invitrogen, Carlsbad, CA) for 1 h and mounted in fluorescent mounting medium (Vector, Burlingame, CA). Ten pancreatic islets from three slides that were obtained from each rat were randomly selected for Olympus confocal microscope (Olympus FluoView 1000, Japan). Three random images from each selected region then were taken, captured, and analyzed using image analysis software (Image-Pro Plus software, Media Cybernetics, Silver Spring, MD). The positive immunolabeled and total cells per high power field in each section were counted, and the percentage of positive labeled cells was calculated.

### 4.6. Terminal Deoxynucleotidyl Transferase dUTP Nick End Labeling (TUNEL)

Apoptotic cell death was determined by TUNEL staining (Roche Diagnostics, Germany) according to the manufacturer’s instructions.

### 4.7. Western Blotting

Pancreatic tissues were harvested in RIPA lysis buffer, and protein concentrations were determined with dichloroacetic acid protein assay kit (Pierce, Rockford, IL). Approximately 50 μg of protein was loaded and separated by SDS-PAGE, transferred to PVDF membrane (Millipore) and incubated with the following primary antibodies: anti-β-actin (Sigma, USA), PCNA (Abcam, UK), anti-Cleaved caspase 3 (Cell Signaling, Danvers, MA, USA), or anti-Bcl-2 (Abcam, UK) at 4 °C overnight, followed by horseradish peroxidase–conjugated IgG as the secondary antibody, and visualized by chemiluminescence.

### 4.8. Enzyme-Linked Immunosorbent Assay (ELISA) Analysis

Pancreatic tissues expression of GLP-1, Insulin, SDF-1, VEGF, TNFα, IL-1β, IL-6, or IL-10 was determined by Quantikine ELISA Kit (R&D Systems, Minneapolis, MN, USA) according to the manufacturer’s instructions.

### 4.9. Statistical Analysis

All experiments were repeated at least three times. All data were expressed as the mean ± SEM. The statistical significances of differences between two groups were determined using Student’s *t*-test (*p* < 0 05). All calculations were performed using the SPSS statistical software (version 13.0, SPSS, Chicago, IL, USA). *p* < 0 05 was considered statistically significant.

## Figures and Tables

**Figure 1 ijms-20-04934-f001:**
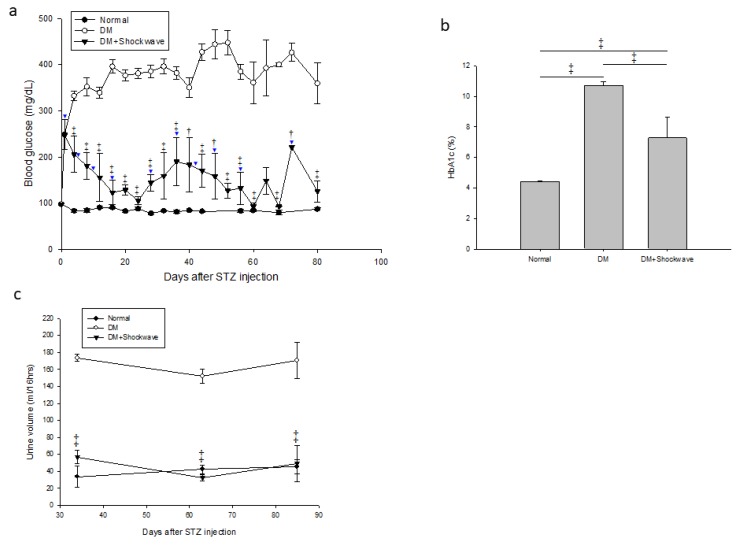
Low-energy SW significantly reduced blood glucose, hemoglobin A1c and urine volume in DM. (**a**) Blood glucose (mg/dL). ^†^
*p* <0.01, ^‡^
*p* <0.001 versus DM. ▼ Low-energy SW; (b) Blood HbA1c. ^‡^
*p* <0.001; (c) Urine volume (ml/16 h). ^‡^
*p* <0.001 versus DM. Data are represented as mean ± SEM (*n* = 12). Normal control (Normal) group, streptozotocin induced diabetes (DM) group and DM with low- energy SW (DM + Shockwave) group, streptozotocin (STZ).

**Figure 2 ijms-20-04934-f002:**
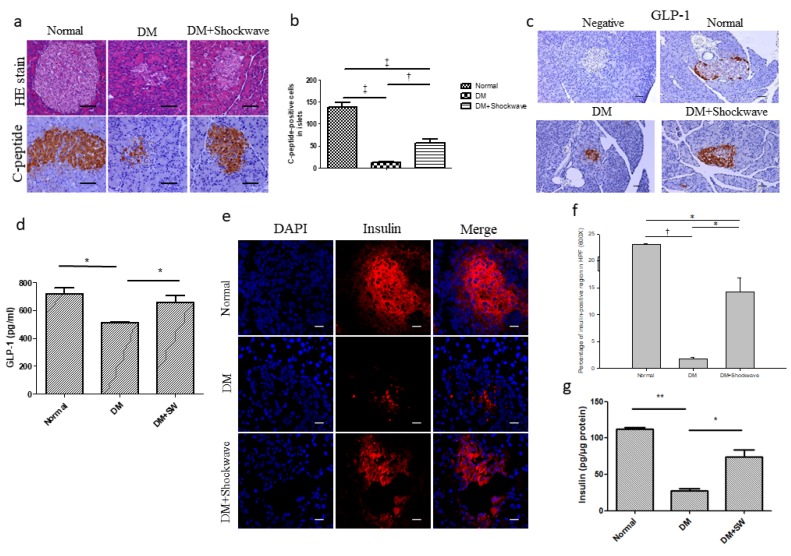
Low-energy SW significantly enhanced pancreatic islets area, c-peptide, GLP-1 and insulin production. (**a**) Representative images of pancreatic tissue stained with HE indicating the pancreatic islets area and IHC detection for C-peptide. Scale bar = 100 μm; (**b**) C-peptide used as a marker of insulin release and quantification of IHC staining C-peptide by image analysis; (**c**) Representative images of pancreatic tissue IHC stained with GLP-1. Scale bar = 50 μm; (**d**) Pancreatic tissue expression of GLP-1 indicating beta cells regeneration. DM with low-energy SW (DM + SW) group; (**e,f**) Representative images of pancreatic tissue immunofluorescence (IF) stained with DAPI (blue), Insulin (red) and quantification of IF staining by image analysis. Scale bar = 20μm; (**g**) Pancreatic tissue expression of insulin. ^‡^
*p* <0.001, ^†^
*p* <0.01, ** *p* <0.01, and * *p* <0.05 (*n* = 6–10).

**Figure 3 ijms-20-04934-f003:**
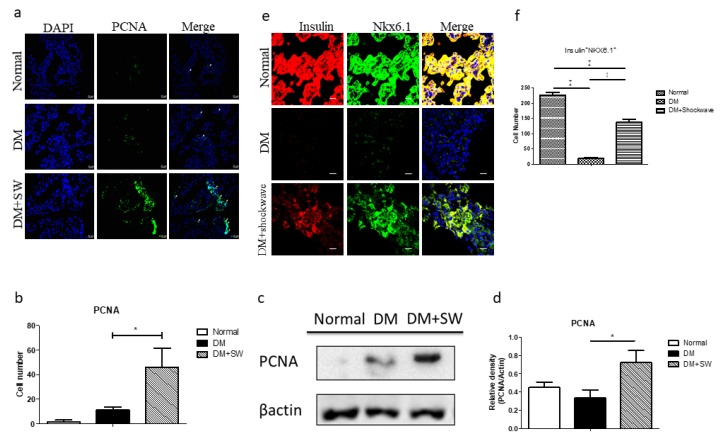
Low-energy SW significantly enhanced proliferation and beta cells regeneration. (**a**,**b**) Representative images of pancreatic tissue IF stained with PCNA indicating cell proliferation, and quantification of IF staining by image analysis. Scale bar = 20 μm; (**c**,**d**) Western blot analysis PCNA expression in pancreatic tissue and quantification of Western blot by densitometric analysis; (**e**,**f**) Representative images of pancreatic tissue IF stained with Insulin (red), Nkx6.1 (green) indicating pancreatic beta cells number and quantification of IF staining by image analysis. Scale bar = 20 μm. * *p* <0.05, ^‡^
*p* <0.001 (*n* = 6–10).

**Figure 4 ijms-20-04934-f004:**
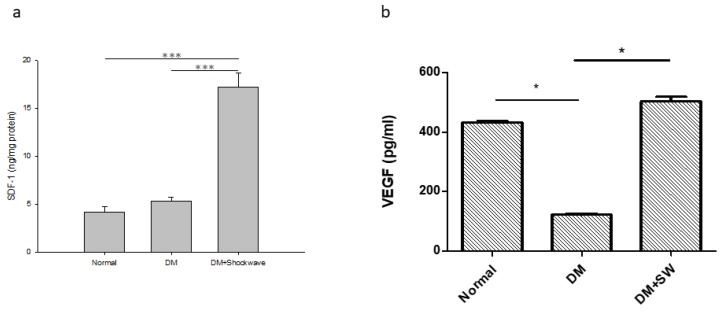
Low-energy SW significantly enhanced tissue repair and angiogenesis. (**a**) Enzyme-linked immunosorbent assay (ELISA) analysis pancreatic tissue SDF-1 indicating tissue repair; (**b**) ELISA analysis pancreatic tissue VEGF indicating angiogenesis. *** *p* <0.001, * *p* <0.05, (*n* = 6–10).

**Figure 5 ijms-20-04934-f005:**
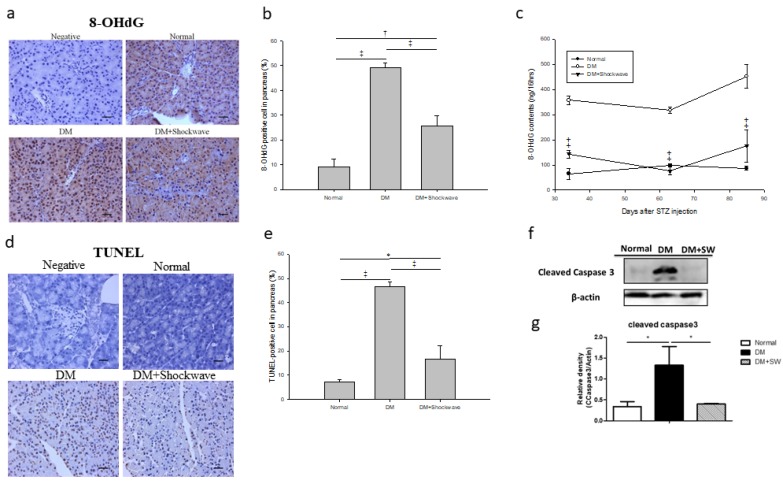
Low-energy SW alleviated oxidative stress, cell apoptosis and enhanced anti-apoptotic. (**a**,**b**) Representative images of pancreatic tissue IHC stained with 8-OHdG indicating oxidative stress and quantification of IHC staining by image analysis. scale bar = 30 μm; (**c**) Urine 8-OHdG (ng/16 h). ^‡^
*p* < 0.001 versus DM. Data are represented as mean ± SEM (*n* = 12); (**d**,**e**) Representative images of pancreatic tissue IHC stained with TUNEL indicating cell apoptosis and quantification of IHC staining by image analysis. scale bar = 30 μm; (**f**,**g**) Western blot analysis cleaved Caspase-3 indicating cell apoptosis in pancreatic tissue and quantification of Western blot by densitometric analysis. ^†^
*p* < 0.01, ^‡^
*p* < 0.001, * *p* < 0.05 (*n* = 6–10).

**Figure 6 ijms-20-04934-f006:**
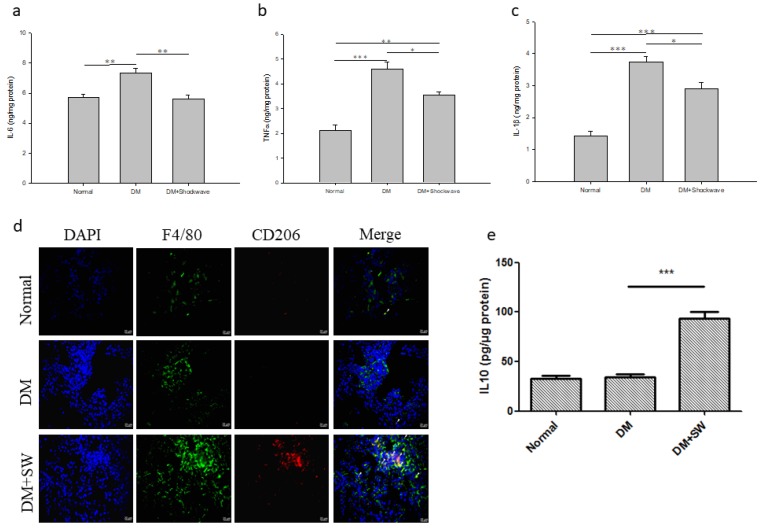
Low-energy SW prevented diabetes-induced pancreatic tissue inflammation. (**a**–**c**) Pancreatic tissue expression of inflammatory mediators IL-6, TNFα and IL-1*β*; (**d**) Representative images of pancreatic tissue IF stained with DAPI (blue), F4/80 (green) and CD206 (red) indicating anti-inflammatory M2 macrophages; (**e**) Pancreatic tissue expression of anti-inflammatory mediator IL-10. Scale bar = 20 μm. * *p* <0.05, ** *p* <0.01, *** *p* <0.001 (*n* = 6–10).

**Figure 7 ijms-20-04934-f007:**
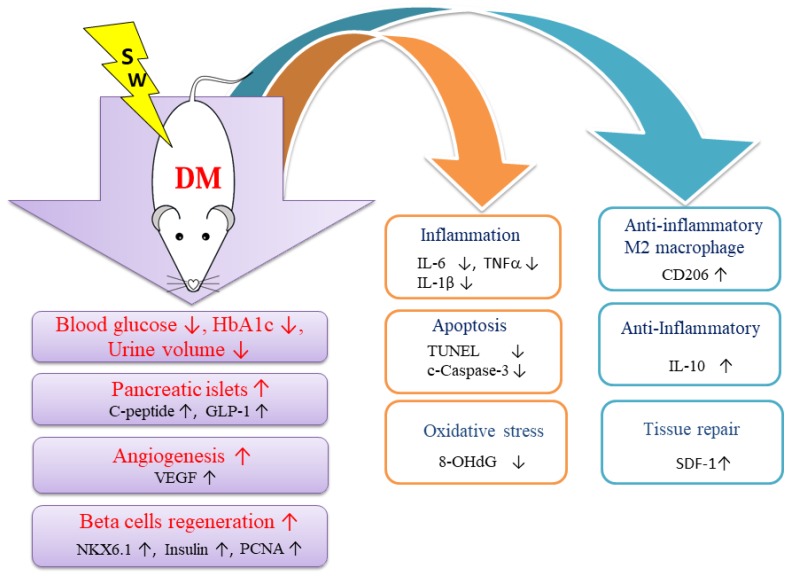
Proposed mechanisms underlying the therapeutic effects of low-energy SW on DM.

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
