# Peer review of "Low-Energy Extracorporeal Shock Wave Ameliorates Streptozotocin Induced Diabetes and Promotes Pancreatic Beta Cells Regeneration in a Rat Model"

_ijms, 2019, doi:10.3390/ijms20194934_

Round 1

Reviewer 1 Report

The authors addressed all of the issues raised in the original review. There is no other concern. 

Author Response

Dear reviewer:

Thanks for your efforts and comments. We have revised the manuscript accordingly. Please see our point to point responses below. 

Comments of reviewer 1:

The authors addressed all of the issues raised in the original review. There is no other concern. 

Response:

Thanks.

We hope that the revised manuscript in its current form is of general interest for the readers and is appropriate for publication in International Journal of Molecular Sciences.

Reviewer 2 Report

The reviewer has read the revised manuscript and authors’ responses. However, most of the responses are insufficient for acceptance. Authors should clearly respond the reviewer’s comments.

Comment #1: Authors changed not only the title but also the conclusion in abstract, as they just demonstrated the preventive effects of low-energy SW on streptozotocin-induced beta cell injury.

Comment #2: The reviewer agrees that low energy SW had therapeutic effects on streptozotocin-induced hyperglycemia. However, the therapeutic mechanisms are questioned. Whereas blood glucose level was increased in non-SW group, it was decreased in SW group immediately after the treatment. This result means that low energy SW prevented acute cytotoxicity of streptozotocin in beta cells. Although authors suggested in the response letter that streptozotocin caused complete eradication of pancreas beta cells, blood glucose level should be never decreased without beta cell preservation. Therefore, low energy SW is considered to prevent the eradication of beta cells. If authors would like to suggest the complete eradication of beta cells in this study, they have to actually demonstrate it in pancreas before the start of SW therapy. Such other potential mechanisms should be discussed in the revised manuscript.

Comment #4: The reviewer’s question was which cells were regenerated. Images in Figure 4 is unclear to show the localization of SDF-1 in pancreas. In addition, whether vWF-positive dot areas were neovessels is unclear in Figure 4.

Comment #5: The reviewer’s question was what types of cells were apoptotic. Authors should show that TUNEL-positive cells were islet cells or endothelial cells. TUNEL-positive cells should be clearly indicated by arrows. In the original comment #5, the reviewer requested additional data of western blotting, including caspase 3 and proapoptotic proteins such as Bax and Bak.

Round 2

Reviewer 2 Report

In the revised manuscript, authors insufficiently responded the reviewer’s comments. The reviewer requested additional data in the original comments.

Comment 1: The reviewer’s suggestion was that in line 29-30, “DM” should be changed into “streptozotocin-induced DM”.

Comment 4: The reviewer’s suggestion was that the images of SDF-1 and vWF immunostaining were unclear. If authors cannot present clearer images, these image data in figure 4 should be removed.

Comment 5: The reviewer requested additional data of western blotting, including caspase 3 and proapoptotic proteins such as Bax and Bak.

Round 3

Reviewer 2 Report

According to Comment #5, apoptotic activity is not always correlated with decreased Bcl2 expression. Thus, it is essential to show the data of both Bcl2 and proapoptotic proteins. If authors disagree to present the data of proapoptotic proteins, they should remove the data of Bcl2 expression.

Author Response

Thanks for your efforts and comments. We have revised the manuscript accordingly. Please see our point to point responses below. 

Comments of reviewer 2:

According to Comment #5, apoptotic activity is not always correlated with decreased Bcl2 expression. Thus, it is essential to show the data of both Bcl2 and proapoptotic proteins. If authors disagree to present the data of proapoptotic proteins, they should remove the data of Bcl2 expression.

Response:

Per the reviewer’s suggestion, we remove the data of Bcl2 expression. We also delete the sentence and word correlated with Bcl2. Please refer to Page 6 - 7, line 138 -156. Line 33, line 177, line 191, and line 207 – 208.